# Neuropsychiatric Disorders in Pediatric Long COVID-19: A Case Series

**DOI:** 10.3390/brainsci12050514

**Published:** 2022-04-19

**Authors:** Rosa Savino, Anna N. Polito, Giulia Arcidiacono, Mariacristina Poliseno, Sergio Lo Caputo

**Affiliations:** 1Department of Woman and Child, Neuropsychiatry for Child and Adolescent Unit, General Hospital “Riuniti” of Foggia, 71122 Foggia, Italy; annanupolito@gmail.com (A.N.P.); arci.giulia@hotmail.it (G.A.); 2Department of Clinical and Experimental Medicine, Infectious Diseases Unit, General Hospital “Riuniti” of Foggia, University of Foggia, 71122 Foggia, Italy; polisenomc@gmail.com (M.P.); sergio.locaputo@unifg.it (S.L.C.)

**Keywords:** long COVID-19, psychosis, movement disorders, neuropsychiatric disorders, COVID-19 haulers, SARS-CoV-2 infection, neuroinflammation, pandemic stress, kynurenines

## Abstract

Few data are available regarding the incidence and the evolution of neuropsychiatric manifestations in children with a history of COVID-19. We herein report five consequent cases of pediatric patients with psychiatric and neurological symptoms of long COVID-19. All patients, mainly males, reported asymptomatic-to-mild COVID-19 and underwent home self-isolation. Abnormal movements, anxiety, and emotional dysregulation were the most recurrent symptoms observed from a few weeks to months after the resolution of the acute infection. A later onset was observed in younger patients. Blood tests and brain imaging resulted in negative results in all subjects; pharmacological and cognitive behavioral therapy was set. A multifactorial etiology could be hypothesized in these cases, as a result of a complex interplay between systemic and brain inflammation and environmental stress in vulnerable individuals. Longer follow-up is required to observe the evolution of neuropsychiatric manifestation in the present cohort and other young patients with previous SARS-CoV-2 infection.

## 1. Introduction

International research has recently focused on the clinical definition of long COVID-19 syndrome, which affects COVID-19 survivors of all ages and all levels of disease severity, and may extend beyond just a few weeks to years [1].

Over 200 symptoms have been attributed to long COVID-19, many of which are nonspecific and highly prevalent in the general population, such as fatigue, sleep disturbance, concentration difficulties, loss of appetite, and muscle or joint pain [2,3]. Convalescence from viral illnesses is a well-known phenomenon and could be characterized by physical, cognitive, emotional, and neurological complications [4]. While ageusia and anosmia are specific and expected COVID-19 related symptoms, fatigue, pain, and emotional liability are much more blurred conditions [5]. The prevalence of symptoms consistent with long COVID-19, including psychosomatic symptoms, have been considerably higher in the pediatric population since the start of the pandemic, and lockdown measures have been shown to have negative effects on the well-being and mental health of children and adolescents [2]. Nevertheless, the neuropsychiatric burden of this pandemic is currently unknown but is likely to be significant since it represents an important cause of morbidity in the pediatric population.

Even though a remarkable incidence of long COVID-19 syndrome has been reported in children, especially in the presence of specific risk factors [6], little data exist in the literature about neuropsychiatric dysfunctions in the evolutionary context of COVID-19, which can potentially relate to an aberrant host response against SARS-CoV-2 infections.

We herein report five consequential cases of pediatric long COVID-19 which showed neurological and psychiatric symptoms in patients with no personal or family history of mental illness.

While evidence suggests that children are less prone to severe disease and death in the acute phase of infection (possibly because of their immune system status with initial relatively low levels of inflammatory cytokine), preliminary data, including this small sample, demonstrate how they can suffer from a wide range of cognitive, affective, behavioral, and perceptual domain [2,7] symptoms after varying periods of later infection.

Moreover, long-term neuro-psychiatric repercussions have been observed in other pandemics before COVID-19, highlighting how large-scale viral infections may cause sustained mental morbidity [7].

Investigating neuropsychiatric sequelae stemming from COVID-19 infection and uncovering underlying pathogenic mechanisms will be critical in developing disease surveillance and evidence-based therapeutic strategies that could target pediatric mental health and preserve children’s neurodevelopment.

## 2. Methods

Starting from 1 February 2021, we identified all patients under 18 years of age with a previous history of COVID-19 (documented with real-time PCR on nasal-pharyngeal swab) consecutively admitted to our unit due to new-onset neuropsychiatric symptoms. A detailed medical history was obtained from their parents. Electroencephalography (EEG), neuroimaging study with Magnetic Resonance Imaging (MRI), and blood analysis were performed. All patients’ caregivers provided informed written consent for publication of their cases. Formal statistical tests were not completed because of the small sample size; the information presented is meant to be descriptive. Table 1 shows demographic information and symptoms of patients in our clinic, while Table 2 shows laboratory data and Table 3 clinical data of study population.

## 3. Case Description

### 3.1. Case 1

A 15-year-old Caucasian male was admitted to our unit in February 2021 with a five-day history of acute psychotic state.

A month earlier, the patient had tested positive for SARS-CoV-2 PCR on a nasopharyngeal swab performed for the onset of fever and anosmia. He was prescribed oral steroids and antibiotic treatment and advised to undergo home isolation.

About 3 weeks after COVID-19 diagnosis, a complete resolution of physical symptoms was noticed. At the same time, his parents referred to a sudden and progressive change in his behavior characterized by restlessness, insomnia, and prominent paranoid delirium, including feelings of concern and guilt.

At admission, his vital signs and neurological examination were normal. A positive SARS-CoV-2 PCR on a nasopharyngeal swab was detected, although a chest X-ray was negative for SARS-CoV-2 pneumonia.

During a medical interview, the patient appeared poorly cooperative and showed partial compromission of time and space orientation, incoherent speech, and a tangential thought process with content notable for religious delusions and confabulation. He referred to auditory and visual hallucinations and showed poor insight and judgment regarding his condition. His recent memory was impaired, in the absence of gross cognitive decline.

No significant abnormalities were detected with either blood tests or in a brain CT, EEG, or MRI, except for a slight descent of the right cerebellar tonsil and a thinning corpus callosum. The probability of encephalopathy was considered low and lumbar puncture was consequently not performed.

The patient started a 1week course of lorazepam in two daily doses of 1 mg, with paliperidone progressively titled to 9 mg/day and nightly lithium sulfate added for the persistence of dysphoria and sudden mood changes.

A significant improvement in religious and paranoid delusions was reported within a week, along with a small improvement in the patient’s insight and judgment. The patient was discharged with negative SARS-CoV-2 PCR and addressed to a close outpatient follow-up 14 days after admission.

Regular examination and evaluation of the patient was periodically carried out, until the last control 11 months later in his COVID-19 recovery. While positive symptoms disappeared completely, negative symptoms persisted. In particular, abulia, alogia, asthenia, social withdrawal, attention deficit, poor language, and anxiety have been detected at all times, albeit they have been progressively milder. EEG, blood count, and ECG were always in the normal range. Lorazepam was discontinued 1 month later upon hospital discharge and lithium after 4 months. Currently, the patient is still on paliperidone and is following cognitive behavioral therapy (CBT).

### 3.2. Case 2

A 7-year-old Caucasian male was admitted to our daily clinic in March 2021 after a new onset of complex motor and vocal tics. His mother reported that these repetitive movements had appeared 1 month earlier, several times a day, especially while he was watching TV or playing with his brother. These movements occurred sequentially, starting with tonic arms extension and applause, and continuing with head extension and the emission of shrill vocalizations.

At the end of January, he tested positive for SARS-CoV-2 PCR on a nasopharyngeal swab and was quarantined with his positive mother for 5 weeks.

Even though mild-severity COVID-19 symptoms were observed, behavioral changes such as hyperactivity, irritability, and insomnia were noticed.

A week after he tested negative, motor and vocal tics appeared, reaching their peak after 2 weeks and then gradually decreasing in intensity and frequency.

At admission, physical and neurologic examinations were normal, although motor restlessness and attention difficulties in assigned tasks emerged. Tics occurred rarely, only during stressful conditions, especially at school.

No abnormalities were detected on EEG, MRI, and cardiological examination. The main laboratory findings were slight leukocytosis, and an increase in erythrocyte sedimentation rate (ESR), CRP, and antistreptolysin titers (AST). The patient tested negative for beta-hemolytic streptococcus on a nasopharyngeal swab. CBT was recommended to manage and treat both tics and cognitive difficulties rather than emotional liability.

### 3.3. Case 3

A 5-year-old Caucasian male was referred in May 2021 due to acute-onset ocular tics.

In this case, the history of COVID-19 dated back to the end of March 2021, when the patient and his mother tested positive for SARS-CoV-2 infection and were advised to self-home isolate for 3 weeks.

A mild severity course of COVID-19 was observed.

A few weeks after having tested negative on a nasal swab, the patient started to suffer from involuntary eye movements such as blinking. Furthermore, his parents described food restriction, separation anxiety, sleep disturbances, and enuresis.

His physical and neurological examinations were normal.

No significant abnormalities were detected with either blood tests, EEG, or MRI. CBT was prescribed, as well as melatonin for sleep problems.

After 3 months, follow-up clinical improvement emerged and ocular tics occurred rarely, especially in the case of fatigue or stress, while enuresis and anxiety significantly improved. The patient is still practicing CBT.

### 3.4. Case 4

A 3-year-old Caucasian female was hospitalized in January 2022 for worsening irritability, tantrum behaviors, and ocular tics.

On April 2021, she was quarantined for 10 days with her mother after testing positive for SARS-CoV-2 PCR on a nasopharyngeal swab. No specific symptoms were referred.

A few months later, she began to show aggressive behavior, separation anxiety from her mother, emotional dysregulation, and, in November 2021, ocular tics.

At admission, her physical examinations were normal, while a neurological investigation confirmed ocular tics (blinking) and a mild language delay. Her brain MRI, awake and sleep EEG, cardiological evaluation, and blood test were normal. Speech and motor therapy were advised and a 3-month follow-up was established.

### 3.5. Case 5

A 2-year 9-month-old Caucasian male was admitted to our unit in February 2022 for acute onset of repetitive and involuntary movements on the face and upper limbs.

In December 2021, he was quarantined with all his family, after they tested positive for SARS-CoV-2 PCR on a nasopharyngeal swab, until early January 2022. Mild flu-like symptoms were reported.

His mother reported that about 3 weeks after a COVID-19 test was negative, the patient started to show abnormal movements such as blinking and myoclonus of upper limbs sometimes associated with staring and grimaces. Irritability, social difficulties, and sleep problems were also detected.

At admission, his physical examinations were normal, but neurological investigation confirmed grimaces and language delay.

Brain MRI, awake and sleep EEG, and cardiological evaluation were normal. The blood test revealed a slight increase in platelets, lymphocytes, and red blood cells, with a mild iron deficiency. Speech and motor therapy were advised and a 3-month follow-up was established.

## 4. Discussion

The COVID-19 pandemic poses a long-lasting challenge, particularly in the pediatric population where the detailed symptomatic manifestations of long COVID-19 remain elusive and may involve a wide range of phenotypes.

The prevalence of long COVID-19 symptoms varied considerably between studies from 4 to 66% [2,8]. An Italian study including 129 children with previous COVID-19 showed that 58% of patients continued to have symptoms for at least 4 months [9]. The most common neuropsychiatric symptoms were insomnia (18%), fatigue (11%), and concentration issues (10%) [6,9]. An Australian analysis suggested that only 8% of children had ongoing symptoms 3–6 months after mild SARS-CoV-2 infection [10]. However, in the UK, <1% of children <17 years old self-reported symptoms of long COVID-19 [11,12]. No gender difference was observed in the prevalence of long COVID-19 in this population [13]. More recently, in a case series of nine patients (<21 years of age) symptoms commonly reported were fatigue (8 of 9 patients), headaches (6 of 9), difficulty with schoolwork (6 of 8), “brain fog” (4 of 9), and dizziness/lightheadedness (4 of 9). Lower scores in sustained auditory and divided attention were also reported in neuropsychological tests [14].

This case series describes the onset and the clinical features of neuropsychiatric post-COVID-19 symptoms in five Italian children.

For our observations, although limited by the small number of subjects involved, a wide range of age involvement (between 2 and 15 years) and a higher prevalence of male subjects were noticed, conversely to what has been observed in previous studies.

Asymptomatic-to-mild COVID-19 was observed in all patients. Hospitalization was not required for any of the cases during acute illness.

The subsequent occurrence of neuropsychiatric complications was noticed at different points of time. Interestingly, in older patients, psychiatric symptoms arose during the acute phase of infection, while in younger patients they occurred after weeks or months.

The most recurrent symptoms reported were abnormal movements, anxiety, and emotional dysregulation. Unusually, involuntary movements were frequently reported in three out of five patients, especially in the younger ones. No previous studies described movement disorders in COVID-19 haulers.

Several theories have been formulated to explain the correlation between COVID-19 infection and the occurrence of psychiatric/neurologic disorders during both the acute and post-acute phase, including direct brain infection, neuroinflammation, autoimmunity, and response to pandemic-related stress [15] (Figure 1).

The functional receptor by which SARS-CoV-2 enters host cells is the ACE2 receptor, which is largely expressed in brain endothelium, neurons, and glia [15,16,17]. The virus possibly infects these cells in the CNS by crossing the blood–brain barrier through retrograde axonal transport from peripheral nerves invasion (such as the olfactory nerve) or hematogenous spread, in more severe cases [18,19]. After neuronal infection, the virions may be released and transneuronal spread occurs to adjacent or pre-synaptic neurons [19].

Several shreds of evidence suggest that neuropsychiatric disorders in COVID-19 patients may be associated with a higher level of pro-inflammatory cytokines, such as IL-6, IL-2, IL-17, granulocyte-colony stimulating factor, and TNF, secondary to immune host anti-viral response [20]. Even in the absence of SARS-CoV-2 infiltration into the CNS, peripheral cytokines may elicit neuropsychiatric symptoms by precipitating neuroinflammatory responses and/or compromised blood–brain interface (BBI) integrity, leading to peripheral immune cell transmigration into the CNS, microglial activation, and disruption of neurotransmission [20,21]. In particular, exposure to pro-inflammatory cytokines has been associated with altered GABAergic transmission in the basal ganglia. Other pro-inflammatory mediators such as interferon-alpha are considered to be associated with a hypo-dopaminergic state in the basal ganglia, which is considered to be a potential inducing factor for psychiatric disorders [19], especially movements disorders.

There is also evidence suggesting that leukocytes, macrophages, and neurons can remain persistently infected by SARS-CoV-2 [22]. Thus, the time course over which infected immune cells could serve as a potential source of neuroinflammation could be significantly longer than the initial infection and acute symptom presentation.

Another pathophysiological mechanism potentially involved in psychiatric symptomatology is the immune dyshomeostasis caused by a novel virus. Aberrant inflammation response, persistent reservoirs of SARS-CoV-2 in specific tissues that trigger post-infection morbidity, re-activation of pathogens, host-microbiome alterations, and molecular mimicry between SARS-CoV-2 and self-proteins have all been thought to play a role in long COVID-19 etiology [3,14,23]. In susceptible individuals (including those with allergies or with familiarity with autoimmune diseases), SARS-CoV-2 could stimulate an aberrant immune response which leads to downstream autoimmunity with secondary damage to the nervous system. Viral infections may precede the development of autoimmunity, creating an inflammatory milieu that favors and promotes the “molecular mimicry” phenomenon through the expansion of host antibodies or lymphocytes, which are cross-reactive both with viral antigen and self-antigen [15].

Osmanov and colleagues suggested that, in children, COVID-19 sequelae could be linked with the mast cell activation syndrome and the Th-2 biased immunological response in children with allergic diseases [24].

Similar to the Epstein-Barr virus, cytomegalovirus, and human immunodeficiency virus, COVID-19 can generate a wide range of autoantibodies, involved in several immunological activities. More than 15 types of antibodies, linked to over 10 autoimmune diseases, have been demonstrated in susceptible individuals with COVID-19 [3,24]. Chronic inflammation may support autoantibodies production, through the alteration of microbiome/virome communities [15].

SARS-CoV-2 nucleic acids and proteins were indeed discovered in the gut of asymptomatic patients at 4 months post-disease onset, leading theoretically to neuroinflammation via the microbiota–gut–brain axis, and in turn to the alteration of neurotransmitter circuitries with secondary neurological symptoms of long COVID-19 [25].

Neuroimaging studies demonstrated structural and metabolic abnormalities in the brain of COVID-19 survivors, which correlated with persistent memory loss, anosmia, fatigue, and headaches. In particular, children with long COVID-19 exhibited hypometabolism in bilateral medial lobes (Amygdalus, parahippocampal gyrus), brainstem, and cerebellum, despite a lower initial severity at the acute stage of the infection in patients analyzed [26].

The effect of COVID-19 on the brain is also associated with the hyperstimulation of the hypothalamic–pituitary–adrenal axis secondary to excessive physiological and psychological stress [16]. Beyond the risks strictly related to the infection, coping with the pandemic situation is a significant psychiatric stressor, especially for vulnerable individuals such as children and adolescents, as their nervous, endocrine systems, and hypothalamic–pituitary–adrenal axes are not well developed [27]. This environmental condition may promote the onset and worsening of neuropsychiatric problems.

Overactivation of the HPA axis leads to secondary production of glucocorticoids with interference with brain metabolism [15,17].

Glucocorticoids affect almost every immune cell type, due to the ubiquitous expression of the glucocorticoid receptor. Within the CNS microglia (a primary target for glucocorticoids) could mediate damage to stress-sensitive regions such as the prefrontal cortex and hippocampus, leading to negative and cognitive symptoms. Glucocorticoids also stimulate neurons to increase glutamate release, which in turn results in further microglial proliferation via the activation of N-methyl-d-aspartate receptors (NMDAR) [28]. In this way, dysregulation of the hippocampus and HPA axis was hypothesized to act synergistically, and activation of the HPA axis was asserted to stimulate the subcortical dopamine system, leading to the development of psychotic symptoms.

Some insights for fine molecular mechanisms underlying long COVID-19 derive from metabolic studies, e.g., an altered tryptophan absorption and metabolism [29].

The etiological link between infection, psychosocial stress, and neuropsychiatric symptoms could be found in the kynurenine pathway, the major route for the catabolism of tryptophan (TRP).

Pro-inflammatory cytokines and adrenal cortisol hormones activate the first rate-limiting KP enzymes, including IDO1 and tryptophan-2,3-dioxygenase (TDO), both peripherally and centrally, shifting tryptophan catabolism towards kynurenines synthesis with secondary depletion of serotonin and melatonin [30]. Melatonin deficiency would cause insomnia or sleep disturbance; instead, a serotonin decrease in SNC could be associated with anxiety and depressive-like behavior.

Thomas et al. found lower levels of tryptophan, serotonin, and indole pyruvate in the plasma of COVID-19 positive patients, as well as higher levels of kynurenine, kynurenic acid, picolinic acid, and nicotinic acid, all of which were positively correlated with interleukin-6 levels [31]. In a separate study, Lionetto et al. [32] demonstrated higher Kyn: Try ratios in the serum of SARS-CoV-2-positive group, which reflects KP activation.

To support the hypothesis of causality between the acute onset of neuropsychiatric symptoms and SARS-CoV-2 infection, it could be considered the clinical entity of pediatric acute-onset neuropsychiatric syndrome (PANS).

PANS, which includes PANDAS (pediatric autoimmune neuropsychiatric disorders associated with streptococcal infections), is supposed to be a heterogeneous disorder in which the major feature is an abrupt onset of obsessive–compulsive disorder (OCD) or restricted eating with at least two concomitant cognitive, behavioral, or affective symptoms such as tics, sleep disruptions, urinary symptoms, anxiety, irritability, or depression [33].

Although PANS is still regarded as a controversial clinical entity in the scientific community [34], the rationale under this disorder is that the initiation of the symptomatic state results from an event (e.g., Group A streptococcal infection) causing a localized immune response in the central nervous system (CNS), and the chronic relapsing course is due to the persistence of the immunological imbalance even after the resolution of the acute phase of the infection [35,36,37].

In this abrupt onset of OCD, various etiological agents, including viruses, mycoplasma pneumonia, and hemophilus influenza, hypothetically act as triggers for immune response, with the consequent release of cytokines in CNS [35]. Remarkably, two Italian cases of new-onset PANS, 2 weeks after a diagnosis of COVID-19, have been reported. Both adolescent patients were referred with an acute onset of motor and vocal tics, as well as behavioral changes [38].

It should be noted that anxiety, attention, and mood concerns observed in our patients could be not directly related to disease mechanisms but to stressors related to COVID-19, such as isolation and virtual schooling, which recurred in all cases.

A pandemic may be the main trigger for the beginning or aggravation of some harmful psychological characteristics, which may lead to behavioral/mental symptoms that in turn increase individual susceptibility to COVID-19 and aggravate the severity of COVID-19 [19].

It is known that acute diseases may trigger the development of somatoform symptoms in predisposed subjects, frequently mislabeled as chronic medical conditions [2,39].

Recent data suggest that 0.9 to 4% of individuals infected with SARS-CoV-2 develop psychotic spectrum disorders [22], both during acute viral infection or after variable periods post-infection, in line with what has been observed in the course of past respiratory viral pandemics [40].

Primary psychosis in the course of COVID-19 has been associated with (i) the absence of predisposing factors such as a family history of severe mental illness or substance abuse; (ii) atypical age of debut; (iii) sub-acute onset of psychotic symptoms (less than 1 week) and fast recovery (maximum 2 weeks) on low antipsychotic doses; and (iv) presence of confusion mixed with the typical psychotic symptoms that allow differentiating this issue from secondary psychotic episodes [41].

A relevant increase in the incidence of acute onset tic-like behaviors, mainly functional, has been documented during the pandemic. In the USA, a tertiary care movement disorders clinic reported an exponential growth of new diagnoses of functional movement disorders of 90.1% in the pediatric cohort [42]. Similar trends were highlighted across Canada, the United States, Europe, and Australia [43].

Even though the etiology of these disorders could be not ascertained, we suppose that it could be multifactorial, as a result of a complex interplay between systemic and brain inflammation and environmental stress in vulnerable individuals.

## 5. Long COVID-19 and Vaccines

The effectiveness of vaccines against long-term symptoms and the incidence of long COVID-19 among pediatric vaccinated people are still unclear.

Movement disorders as a side effect after COVID-19 vaccination are rare, occurring with a frequency of 0.00002–0.0002 depending on the product used, mostly manifesting with tremor [43].

COVID-19 vaccines are estimated to reduce the rate of late complications of COVID-19 disease, mainly by reducing infection rates. A recent and comprehensive case-control study conducted in the UK showed not only a decreased complication rate, but also a reduction of risk of long COVID-19 by around 50% in those who were double vaccinated [6,44]. A mathematical model has shown that including adolescents and children in the vaccination program could reduce overall COVID-19-related mortality by 57%, and reduce cases of long COVID-19 by 75% [45,46].

## 6. Conclusions

Children with long COVID-19 may complain of a variety of neuropsychiatric symptoms that impact their overall quality of life. Identifying early neuropsychiatric symptoms associated with COVID-19 and their pathogenic mechanisms is essential to guarantee a more efficient therapeutic approach and reduce the risk of recurrence and transition to long-lasting disorders.

Dealing with SARS-CoV-2 convalescent patients, who complain of long-lasting subjective physical symptoms, emotional distress should be suspected and a multidisciplinary evaluation should be considered [14], to avoid the risk of an incongruous medicalization and neuropsychiatric labeling [2].

Long-term follow-up is necessary to define the future incidence of neuropsychiatric morbidity after SARS-CoV-2 infection, and prospective studies should be conducted to find out the real impact of COVID-19 on psychopathology.

## 7. Limitations

There are several limitations to our study, including its retrospective and descriptive nature. Given the small sample size, statistical analyses could not be performed. Patients were self-referred to our clinic, which may have resulted in selection bias. We were unable to estimate the prevalence of developing post-acute/long COVID-19 in children after acute infection because we only had data available for those presenting to our clinic.

## Figures and Tables

**Figure 1 brainsci-12-00514-f001:**
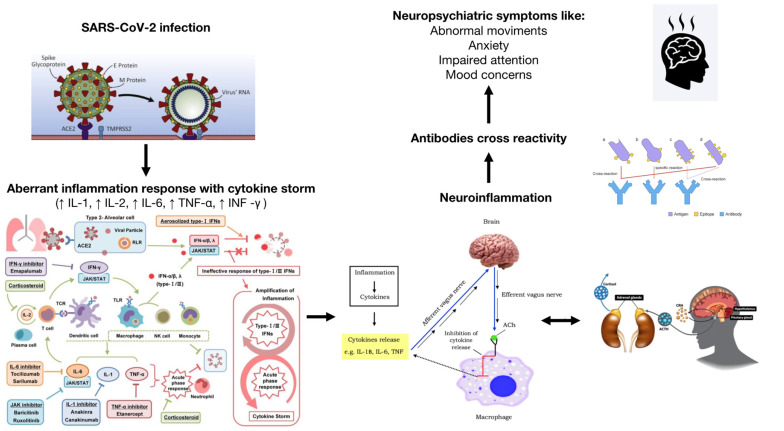
Hypothesized mechanisms of COVID-19 neuropsychiatric symptoms.

**Table 1 brainsci-12-00514-t001:** Demographic information and neuropsychiatric post-COVID-19 symptoms in the study population.

Cases	Sex	Age	COVID-19 Hospitalization	COVID-19 Symptoms	NPI Symptoms	Onset after COVID-19 Disease	Follow-Up and Residual Symptoms
**Case 1** (February 2021)	Male	15	No	Fever and anosmia	Acute psychotic state	3 weeks	12 months after his COVID-19 recovery: asthenia, attention deficit, poor language
**Case 2** (March 2021)	Male	7	No	Mild nasal congestion	Complex motor and vocal tics	1 week	/
**Case 3** (May 2021)	Male	5	No	Only 1-day fever, nasal congestion, and asthenia	Acute-onset ocular tics, food restriction, separation anxiety, sleep disturbances, enuresis	Few weeks	3 months after: rarely ocular tics, occasional enuresis, and anxiety
**Case 4** (January 2022)	Female	3	No	No specific symptoms	Aggressive behavior, separation anxiety, ocular tics	Few months	3-month follow-up
**Case 5** (February 2022)	Male	2	No	Fever and nasal congestion, cough	Involuntary ocular movements, grimaces, upper limbs myoclonus	4 weeks	/

**Table 2 brainsci-12-00514-t002:** Laboratory data of study population.

	Case 1	Case 2	Case 3	Case 4	Case 5
**Blood count ^1^ with leukocyte count ^2^**	Neutrophils: 39.60%; Eosinophils: 5.9%	MPV: 12.7 fl; PDW: 18 fl; Monocytes: 11.5%	Normal	Hemoglobin: 10.9 mg/dL; MCH: 24.8 pg; Hematocrit 32.9% Platelets: 601 × 10^3^/uL; PCT: 0.62% Lymphocytes 56%	RBC: 5.88 10^6^/uL; Hemoglobin: 10.0 mg/dL; MCH: 17 pg; Hematocrit 31.60%; MCV 53.70 fl; MCH 17.00 pg. Platelets: 458 × 10^3^/uL; PCT: 0.45% Lymphocytes 61.10%
**Electrolytes**	Normal	Normal	Normal	Normal	Normal
**Prothrombin Time (PT) (s/INR)**	Normal	Normal	Normal	Normal	Normal
**Activated partial thromboplastin time (APTT) (s/RATIO)**	Normal	Normal	Normal	Normal	Normal
**Fibrinogen (200–450 mg/dL)**	223 mg/dL	203 mg/dL	217 mg/dL	**475 mg/dL**	**257mg/dL**
**Thyroid function indices (ATM, ATG, TSH, FT3, FT4, Thyroglobulin)**	FT3 5.68 pg/mL (n.r.2.50–5.20 pg/mL)	Normal	Normal	Normal	Normal
**Ceruloplasmin (22–58 mg/dL)**	**16 mg/dL**	**19 mg/dL**	35 mg/dL	/	/
**Copper (65–165 µg/dL)**	**56 µg/dL**	66 μg/dL	74 μg/dL	/	/
**Transaminases (ALT, AST, GGTP)**	Normal	Normal	Normal	Normal	Normal
**Serum Creatinin +** **Azotemia**	Normal	Normal	Normal	Normal	Normal
**Chemical physical examination of urine and urinary sediment**	Specific weight: 1028 (n.r. 1011–1025)	Normal	Normal	Specific weight: 1006 (n.r. 1011–1025)	Specific weight: 1010 (n.r. 1011–1025)
**Serum Iron (60–160 ug/dL)**	60 ug/dL	68 ug/dL	76 ug/dL	53ug/dL	55 ug/dL
**AST (children up to 150 U.I/mL)**	116 U.I/mL	**161 U.I/mL**	135 U.I/mL	5 U.I/mL	/
**CRP (0.00–2.00 mg/L)**	1.2 mg/L	**2.5 mg/L**	0.5 mg/L	**27.9 mg/L**	0.9 mg/L
**ESR (1 h 1–16)**	9 mm/h	17 mm/h	2 mm/h	**56 mm/h**	/
**Cortisol (4.3–27 µg/dL h. 7–9)**	28.56 µg/dL	7.09 µg/dL	9 µg/dL	/	/

^1^ Blood count: White Globules, Red Globules, Hemoglobin, Hematocrit, Average Globular Value, MCH, MCHC, RDW, Plates, MPV, PCT, PDW. ^2^ Leukocyte count: Neutrophils, Lymphocytes, Monocytes, Eosinophils, Basophils/= insufficient blood sample.

**Table 3 brainsci-12-00514-t003:** Comparison of the five patients’ clinical data.

	Case 1	Case 2	Case 3	Case 4	Case 5
EEG	Normal	Normal	Normal	Normal	Normal
MRI	Slight descent of the right cerebellar tonsil and thinning corpus callosum.	No abnormalities found	No abnormalities found	No abnormalities found	No abnormalities found
ECG	Sinus Tachycardia at 102 bpm. QT 312 ms, QTc 410 ms.	Normal sinus rhythm. QT 336 ms, QTc 425 ms.	Respiratory sinus arrhythmia at 76 bpm.	Normal sinus rhythm at 106 bpm.	Respiratory sinus arrhythmia at 75 bpm.
Therapy	Drugs Paliperidone 9 mg/die (ongoing); Lorazepam 1 mg × 2/die (discontinued after 1 month) Lithium Sulfate 83 mg/die (discontinued after 4 months) CBT	CBT	Melatonin (1 mg/die) CBT	CBT + Speech therapy	CBT + Speech therapy

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
