# Peer review of "Neuropsychiatric Disorders in Pediatric Long COVID-19: A Case Series"

_brainsci, 2022, doi:10.3390/brainsci12050514_

Round 1
Reviewer 1 Report
Dear authors
Cases of neurological disorders due to alterations in the body's metabolic homeostasis, which leads to dysregulated biochemical signaling in neuronal inputs, especially in the circuits that associate the limbic system with the frontal cortex. These deregulations, whether at a molecular or structural level, as can be seen in regions associated with the limbic system, may be a plausible explanation for the occurrence of neurological and neuropsychiatric disorders associated with Sars Cov-2.
Neuroinflammation due to cytokine storm, for example high concentration of IL-1, IL-6 and TNF in receptors present in the central nervous system, focusing on DLPFC, predispose these cases, bringing the precepts of biological plausibility.
The series of cases raised by the authors bring clarity and give us a direction to these possible associations.
I give a favorable opinion to the manuscript.
Minor comments: Insert the neuroimages of the patients and, if possible, bring in tables the descriptive measures of all the variables analyzed, if possible.
Author Response
Dear Author,
please see the attachment

Reviewer 2 Report
In this report, the authors describe a pediatric cohort with a recent Sars-CoV-2 infection presenting to their Department for neuropsychiatric symptoms in order to analyze the possibile impact of this agent in the clinical reported features.
Although interesting, the absence of mathed control subjects and the limited sample size not allowing any formal statistic, it is very difficult to draw any other assumption to high risk of biased report.
In particolar, it is well known that convalescence after a viral infection may last longer than usual (https://pubmed.ncbi.nlm.nih.gov/29237442/) and that when dealing with SARS-CoV-2 convalescent patients, who complain long-lasting physical symptoms, emotional distress should be suspected and a multidisciplinary evaluation should be considered (https://pubmed.ncbi.nlm.nih.gov/34250520/). In addiction, since any comparison is allowed, any assumption is possible about Sars-CoV-2 eventual causative role.
For this reason, I suggest limiting the report to neurologic clinical features (and not psycological one, with term like “emoziona lability”), to limit biased results.
Moreover, a table comparing patients with the reported features in comparison with other patients in the same period with simular features without Sars-CoV-2 and with reported diagnostic tests might be useful for the scientific reader.
In addiction:
Clinical picture is well presented for all patients. A Figure showing symtoms presentation over time might be welocomed. Include diagnostic test in the table. Any genetic test performer for any patient?
Please be more balanced in the discussion, the risk of incongruous medicalization and medical labeling in long lasting symtoms is very high (https://pubmed.ncbi.nlm.nih.gov/31738390/) and be aware that acute diseases may trigger the development of somatoform symptoms in predisposed subjects.
PANS entity is controversial (https://pubmed.ncbi.nlm.nih.gov/30996598/) and not accepted by all the medical literature. I suggest to replace with description of the clinical course such as “abrupt onset of OCD”.
Causation explaination should be limited, since any control match has been set in your paper.
Reviewer 3 Report
Summary: COVID-19 infections can result in delayed neuropsychiatric manifestations both in adults and children. This is still a very understudied topic but has huge implications. In the present study, the authors have documented their observations of 5 pediatric cases where COVID-19 infections resulted in several neuropsychiatric symptoms including problems with movements, anxiety, and emotional dysregulation. The study has not done a thorough analysis using a big sample but it still seems to be an important documentation at this stage of the pandemic. While it is possible that these behavioral symptoms are resulting from an interaction between systemic and brain inflammation combined with environmental stressors in the form of social isolation, it cannot be ruled out that they are merely a result of one of these factors. Thus, this study can provide clinicians with an idea of testing some of these anxiety-drugs during the isolation period and having a more controlled analysis of emergence of the neuropsychiatric symptoms. I don't have any major criticism of the study.The manuscript is nicely written and language is very clear. The methods and other information are in good detail. I am just wondering if case number 4 and 5 showed any improvements similar to the other three.
Minor: 1) 177 line - in "any case" replace with in "no case". 2) Put the full form of PANS on its first occurence.
Author Response
Reviewer #3:
I don't have any major criticism of the study. The manuscript is nicely written and language is very clear. The methods and other information are in good detail. I am just wondering if case number 4 and 5 showed any improvements similar to the other three.
Minor: 1) 177 line - in "any case" replace with in "no case". 2) Put the full form of PANS on its first occurence.
Authors’ Response
Dear reviewer, thank you for your positive opinion and for the relevant comments you have addressed to us. As you suggested we provided to correct “any case” with “ no case”, and we put the integral form of PANS at its first occurrence.
With regard the 4 and 5° case, the evolution of neuropsychiatric symptoms remains currently unknown, as 3-months follow-up is still ongoing.
Reviewer 4 Report
I would like to thank the authors for taking the time to conduct this research and writing this manuscript about such an important topic as neuropsychiatric disorders in pediatric patients with long COVID-19. In light of the unprecedented times we are living, this is of utmost interest and importance. This manuscript covers a rather novel issue affecting young children, the side effects of COVID-19 infection in the long term. I would like to acknowledge all the time and effort authors have put into this work, as I believe this research is relevant to the field of child development and behavior. Even though I believe that this manuscript presents sound research, I think that it could be improved by making some changes.
First, I would suggest providing more conceptual information in the introduction and explaining more explicitly why a case series of young patients affected by COVID-19 and the neuropsychiatric disorders seemingly triggered by the infection is relevant to the field. Second, I would also suggest reviewing the discussion, for instance, authors refer in lines 186-188 to several theories that have been formulated to explain the possible relationship between Sars-Cov-2 infection and the occurrence of psychiatric/neurologic disorders, but yet they don’t mention any of these theories, leaving the reader wondering about such theories. I believe that by elaborating on this, readers will have a better understanding as to why mentioning these theories is important to this study. The third point I would like to bring up is the terminology used to refer to the disease -throughout the manuscript, it is referred to as COVID, COVID-19, Sars-Cov-2, and I would suggest sticking to one term when referring to the disease for fluency and coherence purposes. The last point I would suggest reviewing is language use. I believe that the manuscript would be significantly improved by going over minor editorial aspects, such as writing mechanics and paragraph structure. The organization and development of ideas are good but can be improved. I suggest revising it by paying close attention to sentence structure, punctuation marks, and word choice.
Author Response
Authors’ Response
Dear Reviewer, thank you for your precious observations.
Both the “Introduction” and “Discussion” section were modified in order to better explain the relevance of our case series from a conceptual point of view. We have also deeper explained the theories that are mentioned in Discussion regarding the relationship between Sars-CoV2 infection and the occurrence of psychiatric/neurologic disorders.
Finally, all manuscript underwent thorough English language revision to ensure fluidity and consistency throughout the text with special attention to the terminology used to refer to COVID-19.
Round 2
Reviewer 2 Report
The paper overall quality is improved. Issues previously raised have been addressed as possible. In case other modifications were requested by other reviewers, it seems reasonable to shorten your paper in the "discussion" section as it is hard to follow in that part.